# Selection rules for all-optical magnetic recording in iron garnet

A. Stupakiewicz[1], K. Szerenos[1,2], M.D. Davydova[3,4], K.A. Zvezdin[3,4], A.K. Zvezdin[3,4], A. Kirilyuk[2,5] & A.V. Kimel[2,6]

Rapid growth of the area of ultrafast magnetism has allowed to achieve a substantial progress in all-optical magnetic recording with femtosecond laser pulses and triggered intense discussions about microscopic mechanisms responsible for this phenomenon. The typically used metallic medium nevertheless considerably limits the applications because of the unavoidable heat dissipation. In contrast, the recently demonstrated photo-magnetic recording in transparent dielectric garnet for all practical purposes is dissipation-free. This discovery raised question about selection rules, i.e. the optimal wavelength and the polarization of light, for such a recording. Here we report the computationally and experimentally identified workspace of parameters allowing photo-magnetic recording in Co-doped iron garnet using femtosecond laser pulses. The revealed selection rules indicate that the excitations responsible for the coupling of light to spins are *d-d* electron transitions in octahedral and tetrahedral Co-sublattices, respectively.

[1] Faculty of Physics, University of Bialystok, 15-245 Bialystok, Poland. [2] Institute for Molecules and Materials, Radboud University, 6525 AJ Nijmegen, The Netherlands. [3] Moscow Institute of Physics and Technology (State University), Dolgoprudny, Russia 141700. [4] Prokhorov General Physics Institute of the Russian Academy of Sciences, Moscow, Russia 119991. [5] FELIX Laboratory, Radboud University, 6525 AJ Nijmegen, The Netherlands. [6] Russian Technological University (MIREA), Moscow, Russia 119454. Correspondence and requests for materials should be addressed to A.S. (email: and@uwb.edu.pl) or to A.V.K. (email: a.kimel@science.ru.nl)

Ultrafast control of the electron spin in the matter with the help of ultrashort light pulses has evolved into one of the hottest topics of the fundamental magnetism and generated significant interest in the area of information technology. In particular, the fact that magnetization can be reversed by femtosecond laser pulses may result in new concepts of storage devices with unprecedented bit rates. Speaking in general, moreover, finding an electronic transition a subtle excitation of which can launch dramatic changes of electric, optical or magnetic properties of media is one of the long-standing dreams in the field of photo-induced phase transitions[1–5]. This field got a great boost due to the development of new femtosecond laser sources allowing to explore photo-induced phase transitions in a broad spectral range and on a time scale of elementary processes in condensed matter such as electron–electron, electron–lattice, spin–lattice interactions, period of oscillations of lattice atoms, or spins. The interest to magnetic photo-induced phase transitions has been continuously fueled due to its relevance to magnetic recording technology and the discovery of the magnetization switching by a femtosecond laser pulse alone[6–10] triggered intense discussions about mechanisms responsible for these laser-induced changes. For a long time practically all research on all-optical magnetic switching was limited to metals and relied on mechanisms directly related to laser-induced heating close to the Curie temperature. Ultrafast light–matter interactions in this case are therefore explained with the help of a simplistic temperature model, where features of the electronic structure hardly play any role[9]. However, the most intriguing pathways to change the magnetization with a laser pulse are those that otherwise leave the entropy of the whole system almost intact. Such precisely directed impact has the potential to reduce energy waste and also to provide novel possibilities for ultrafast transfer of recorded data in the optical telecommunication systems using ultrashort laser pulses. Recently it has indeed been shown that using a single 50 fs laser pulse one can permanently switch the magnetization in Co-doped yttrium iron garnet thin film (YIG:Co) with (001) plane of the sample[10]. In this material, the Co-dopants are responsible for strong magnetocrystalline anisotropy, making cube diagonals <111> to be eight easy directions. It was experimentally demonstrated in our previous work[10] that optical excitation of localized $d-d$ transition in Co-ions at 0.95 eV with light polarized along [100] or [010] axis was able to modify the magnetic anisotropy and thus to switch the magnetization between the metastable states. This observation raised questions about selection rules e.g., the feasibility of magnetic recording using light of other photon energies and light polarizations.

In this paper, we demonstrate that contrary to previous reports[10], all-optical magnetic recording under both single-pulse and multiple-pulse sequences can be achieved, in narrow spectral ranges not with one but with two pairs of light polarization, either along [110] and [1−10] or [100] and [010] crystallographic axes of the garnet. Depending on the pump wavelength, different pairs of polarizations must be used to record and erase a magnetic domain. Such a possibility has not been either predicted or anticipated before. Note that the knowledge of the mechanisms, i.e., electronic transitions, responsible for the effect is the main obstacle preventing any quantitative modeling of light–matter interactions. Here, we reveal these transitions and thus open up the way for theoretical and computational studies of all-optical magnetic recording and ultrafast photo-magnetic phase transitions in dielectrics.

## Results

**Model of ultrafast photo-magnetic switching.** Aiming to define the workspace of the key parameters allowing the switching, we developed a phenomenological model of the photo-induced magnetization dynamics in YIG:Co film based on the Landau–Lifshitz–Gilbert (LLG) equation:

$$\dot{\mathbf{M}} = \gamma \left[ \mathbf{M}, \frac{\delta \Phi}{\delta \mathbf{M}} \right] + \frac{\alpha}{M_s} \left[ \mathbf{M}, \dot{\mathbf{M}} \right] + \mathbf{T}_E(\mathbf{M}, \mathbf{E}), \qquad (1)$$

where $\gamma$ is the gyromagnetic ratio, $\alpha$ is the Gilbert damping parameter, $M_S$ is the saturation magnetization, $\mathbf{E}$ is the electric field vector, and $\Phi$ is the thermodynamic potential of the unperturbed magnet in the form of free energy.

The last term in the equation stands for optically induced spin-torque $\mathbf{T}_E(\mathbf{M}, \mathbf{E})$. The latter may be represented in the following way in the Cartesian coordinate system:

$$\mathbf{T}_E(\mathbf{M}, \mathbf{E}) = -\gamma[\mathbf{M}, \mathbf{H}_{eff}(\mathbf{M})], (H_{eff}(\mathbf{M}))_i = -\beta_{ijkl}M_jE_kE_l^* \qquad (2)$$

where $\beta_{ijkl}$ is the photo-magnetic tensor[10]. The number of independent tensor components can be found taking into account the $4mm$ point group for (001) oriented YIG:Co film[11,12] (see Fig. 1a) and the fact that the tensor $\beta_{ijkl}$ must be invariant with respect to permutations of the last two indices. The material parameters and coefficients of the photo-magnetic tensor a, b, and c (see Methods) were obtained by fitting the theoretical results to the experimental data from ref. [6].

There are four easy magnetization axes near <111>− type directions and 8 ground states for magnetization vector, taking into account two possible orientations along each axis. We denote these metastable magnetization states with numbers 1−8 (see in Fig. 1b). Earlier studies allowed to estimate that the life-time of the photo-magnetic anisotropy is about 20 ps[13]. The model accounts for this by introducing a temporal relaxation of the tensor components $\beta_{ijkl}$. Therefore, during the first 20 ps after the optical excitation the motion of spins will mainly be defined by the field of the photo-induced magnetic anisotropy and afterwards the motion will be driven by the effective field of the steady-state magnetocrystalline anisotropy.

We demonstrate the modeling of the photo-magnetic switching between possible magnetization axes (phases), e.g., 1–4, 1–5 and 1–8 (see Fig. 1b). Taking 1 as the initial magnetization state we performed the simulations of the trajectories of the magnetization vector varying the intensity of light and magnetic damping $\alpha$ (see Methods). Figure 1c, d summarizes the phase diagram of the simulations showing the final states to which the magnetization relaxes after being excited by light of two polarizations ($E \parallel$ [100] and $E \parallel$ [110]). The switching can be observed for these polarizations in a wide range of light intensities. This model demonstrates different trajectories of the switching (see Fig. 1e) between: (i) only in-plane magnetization components (between [110] and [1−10] axes, see 1−4 trajectory); (ii) only perpendicular magnetization components (between [001] and [00−1] axes, see 1−5 trajectory); and (iii) simultaneously in-plane and perpendicular magnetization components (between [111] and [1−1−1] axes, see 1−8 trajectory). This model allows to simulate the behavior of a single spin in a crystal lattice with the symmetry of the iron garnet, where the Gilbert damping is taken as a constant. The damping was treated as a fitting parameter. The best agreement of our simulations with the experimental data reported here was achieved for $\alpha = 0.28$. Note that the previously mentioned experimental value of $\alpha = 0.2$[10,13] was measured using the technique of ferromagnetic resonance, where the amplitudes of magnetization oscillations are small. It is well known, however, that the damping increases with the amplitude of the magnetization oscillations[14].

Hence the simulations suggest that in addition to the photo-magnetic recording reported in ref. [10], the symmetry of the garnet

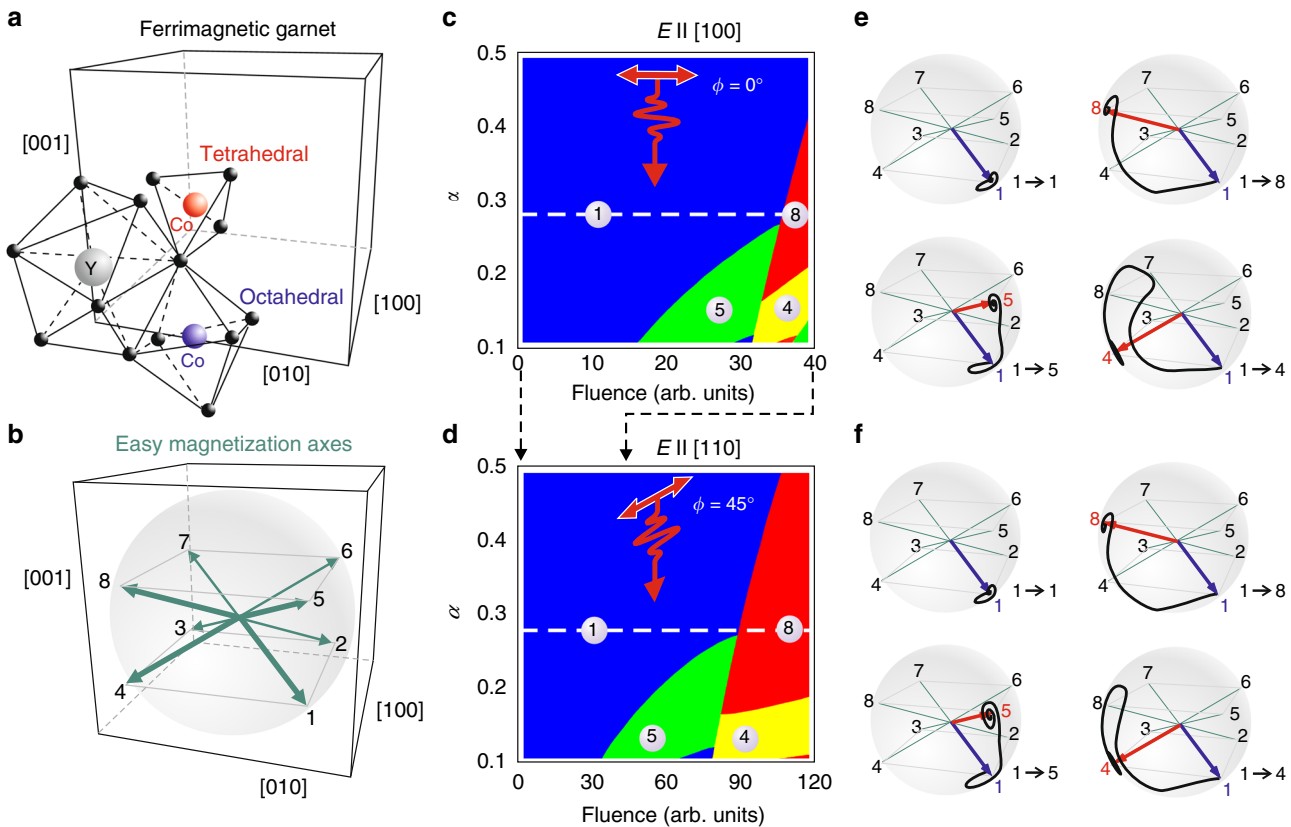

**Fig. 1** Photo-magnetic switching in YIG:Co. **a** Crystal structure of iron garnet. Co-ions can enter the crystal by taking the positions in octahedral and tetrahedral environments. **b** Eight metastable states of the magnetization in relation to the crystal symmetry of iron garnet. **c** Phase diagram of the photo-magnetic switching from state 1 with light polarization $E \parallel [100]$ as a function of Gilbert damping $\alpha$ and light fluence. The white dashed line shows the value $\alpha = 0.28$. **d** Phase diagram of the photo-magnetic switching from state 1 with light polarization $E \parallel [110]$ for as a function of Gilbert damping $\alpha$ and light fluence. The white dashed line shows the value $\alpha = 0.28$. **e** Trajectories of the switching from state 1 to states 4, 5, and 8 using light polarization $E \parallel [100]$. **f** Trajectories of the switching from state 1 to states 4, 5, and 8 using light polarization $E \parallel [110]$. In all shown trajectories the blue arrow is the initial magnetization state and the red arrow is the state after the switching. The trajectories calculations are performed for various damping and fluence, based on the phase diagrams to help illustrate the switching. Trajectories of the switching from state 1 to 8 was performed for $\alpha = 0.28$. The switching paths from 1 to 4 and from 1 to 5 were simulated for $\alpha = 0.12$. Fit parameters for all simulations are $b = -a/3$, $c = a/2$ (see Eq. 4 in Methods)

films supports the possibility of the magnetic switching using other light polarizations than [100] and [010], for which the mechanism of the switching must rely on excitation of other electronic transitions with other selection rules.

Compared to pure YIG, the absorption spectra of YIG:Co in the range of photon energies from 0.5 eV to 2 eV show a number of additional peaks associated with Co-ions[15]. The most pronounced peaks are observed around 1.13 eV (the wavelength is 1.1 μm) and 0.95 eV (the wavelength is 1.3 μm)[16–18]. So far the most conventional way to illustrate the switching is based on scanning a polarized laser beam, consisting of series of femtosecond laser pulses, across the sample surface[6–8]. To verify the feasibility of the magnetic switching at other photon energies and other polarizations of light, we employed this method for two polarizations of light [110] and [1−10]. The experiments were performed at several photon energies in the range from 0.83 eV to 1.24 eV.

**Multiple-pulse magnetic switching**. The initial magnetic domain pattern as observed in a polarized light microscope (see Methods) is shown in Fig. 2. The domain structure in YIG:Co thin films has been studied and reported in detail in refs. [22–24] together with a description of the experimental procedure of defining the magnetization in the domains (see Methods). We rotated

the linear polarization over 45° with respect to [100] axis ($\phi$ is the angle between the electric field of light and [100] axis) and searched for the wavelength allowing the switching. After scanning the beam with $\phi = 45°$ and photon energy 1.1 eV ($\lambda = 1140$ nm) the large black domains are turned into white ones. The orthogonal polarization along [1−10] crystallographic axis does the opposite. Both polarizations are different from those used for the switching in ref. [10] indicating that here it most likely proceeds along different microscopic mechanism, i.e., relies on excitation of different electronic transitions than those reported earlier.

Figure 3 shows how the switched area increases with an increase of number of pump pulses $N$ using quasi-static magneto-optical imaging in Faraday geometry within 10 ms frame after excitation with a single pump pulse (see Methods) (see Fig. 3a). By increasing the pump pulse fluence, the number of pulses required for the switching can be brought down to one. Note that the appearance of a domain after a certain number of pulses is a very reproducible effect that is determined by a balance between the repetition rate of the pulses and magnetostatic field driven domain wall motion (for more details see Methods). The process of domain formation using multiple pulses for 105 mJ cm$^{-2}$ is following: after first pump pulse the magnetization was switched, however the switched area was not stable enough. The next pulses resulted in an expansion and thus stabilization of the recorded

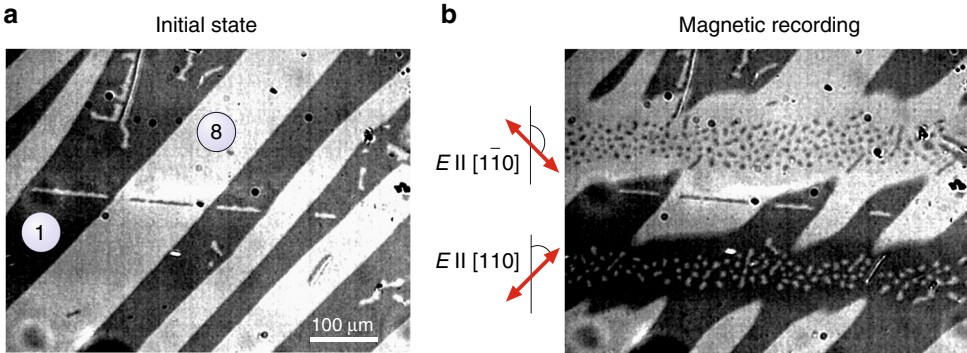

**Fig. 2** Magnetic domains recorded by scanning polarized laser beam. **a** Initial domain pattern in YIG:Co film. Large black and white stripes correspond to the domains with two orientations of the magnetization—1 and 8 magnetization states in Fig. 1b. **b** The result of a scan with $E \parallel [110]$ polarized light ($\phi = 45°$) and $E \parallel [1-10]$ polarized light ($\phi = 135°$). The pump beam was focused to a spot with the radius $r = 65\,\mu m$. The scan velocity was 200 $\mu m\,s^{-1}$ with 1000 pulses per second. The beam contains 50 fs laser pulses with the fluence of 100 mJ cm$^{-2}$. The central photon energy is 1.1 eV ($\lambda = 1140$ nm). The recorded tracks are repeatable and stable for a long time at room temperature without external magnetic field

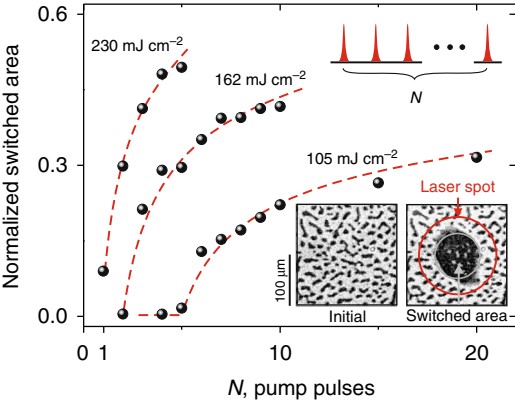

**Fig. 3** Photo-magnetic switching with multiple pulses at different pump fluences. A single pulse with the fluence of 230 mJ cm$^{-2}$ is able to create a stable domain. The time period between the pulses is 1 ms (repetition rate was 1 kHz). The central photon energy of the pump is 1.1 eV ($\lambda = 1140$ nm) and the pump polarization is along [110]. The normalized switched area was calculated as the ratio of the recorded domain area (the black large domain) to the area of pump laser spot $\pi r^2$ (limited by red circle). Dashed lines are guides to the eye. The inset shows the magnetic domain patterns before and after the laser excitation

domain. As a matter of fact, the switched domains have been clearly and reproducibly seen after $N = 5$ (see Figure 3).

**Selectivity of photo-magnetic recording in Co-doped YIG.** Figure 4 shows a summary of the switching efficiencies as functions of light wavelength, polarization angle, and pulse fluence. The spectral dependence shown in Fig. 4a reveals two resonant features around 1.1 eV and 0.95 eV, respectively. Figure 4b shows that the most efficient photo-magnetic switching with photon energy of 0.95 eV is observed for [100] polarization. If the pumping is at photon energy of 1.1 eV, the switching has the maximum efficiency for [110] polarization. This exactly corresponds to the modelling results revealing different threshold intensities for the switching with different light polarizations (see phase diagrams in Fig. 1). Finally, pumping at 0.95 eV is characterized by three times lower threshold intensity compared to the case of pumping at 1.1 eV (Fig. 4c).

The revealed selection rules shown in Fig. 4a, b allow us to identify the microscopic mechanisms responsible for the photo-

magnetic recording. According to refs. [17,18] a sharp peak around 1.1 eV is due to $^4T_1 \rightarrow {}^4T_2$ electronic transition in Co$^{2+}$ ions in octahedral sites. A broader peak at 0.95 eV corresponds to $^5E \rightarrow {}^5T_2$ transitions in Co$^{3+}$ ions and $^4A_2 \rightarrow {}^4T_1$ transitions in Co$^{2+}$ ions in tetrahedral sites[17]. In both experiment and simulations the threshold fluence for the switching with light polarization along [110] axis is about three times larger than the threshold for light polarization along [100] axis (see Figs. 1c, d and 4c). The reason for this is possibly related to the fact that, as a general crystallographic property, the oscillator strength of low-symmetry tetrahedral sites is much stronger than high-symmetry octahedral sites[16,19,20]. As a result, the response of octahedral ions becomes clearly noticeable only when sufficient pump fluence is applied. The contribution to the magnetic anisotropy at room temperature is also stronger from Co-ions in the tetrahedral sites[21].

## Discussion
Finally, it is interesting to note the remarkable efficiency and selectivity of the switching by pumping different electronic transitions in Co-dopants. Apart from purely fundamental interest and impact on further development of theory of photo-magnetism, the discovered multi-wavelength magnetic recording in iron garnet may spur applications of photo-magnetism in information processing technology. Firstly, the possibility of recording at several wavelength shows that the wavelength for the most efficient photo-magnetic recording is not fixed but can be tuned in the range relevant to optical telecommunication systems. Secondly, our work also shows that information transferred from light to spins can be encoded not only in polarization state, but also in the wavelength and the intensity of the recording beams. In fact, our work shows that due to the additional degree of freedom one can increase the intensity of data traffic from light to spins. For instance, different spectral components of light can carry information, which due to dispersion will be focused and recorded on different domains of a storage medium. Thirdly, the discovered selection rules allowed us to reveal the origin of the photo-magnetic centers and thus to perform quantitative estimates of the efficiency of photo-magnetic recording. In the studied films only one of forty Fe ions is substituted by Co[22–24]. It means that single-photon excitation of Co ion must be, in principle, sufficient to control the spins in a rather large volume of 28 nm$^3$ (see Methods). Fourthly, the selection rules reported in this paper reveal the possibility of an independent control of spins coupled to octahedral and tetrahedral photo-magnetic centers. Employing the discovered degrees of freedom, polarization pulse

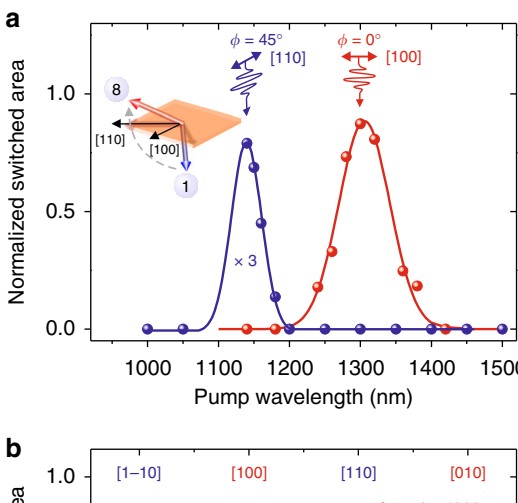

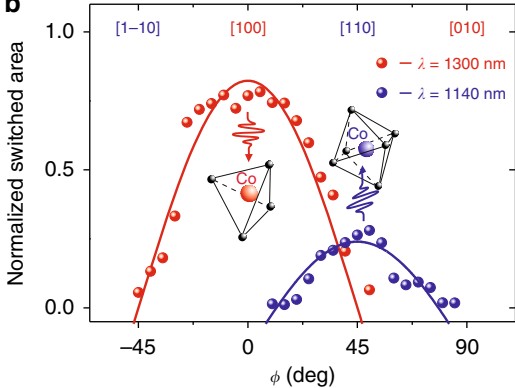

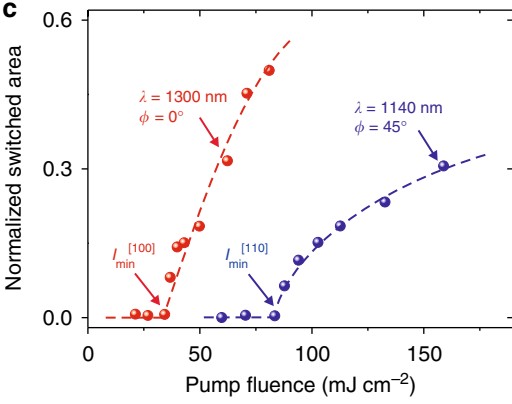

**Fig. 4** Selection rules of photo-magnetic switching on wavelength and polarization. For measurements 5 laser pump pulses ($N = 5$) were used. **a** Normalized switched area as a function of the photon energy for two polarizations $E \parallel [110]$ ($\phi = 45°$) and $E \parallel [100]$ ($\phi = 0°$). Solid lines were fitted by Gaussian function. **b** Normalized switched area as a function of the angle $\phi$ for the incoming polarization for two photon energies 0.95 eV ($\lambda = 1300$ nm) and 1.1 eV ($\lambda = 1140$ nm). Solid lines are guides to the eye. **c** Fluence dependence of the normalized switched area for two photon energies 0.95 eV and 1.1 eV. Dashed lines are guides to the eye. The measurements at different photon energies reveal different threshold fluences $I_{min}$

shaping and the principles of coherent control, one can anticipate new opportunities for ultrafast optical manipulation of coherent spin waves far from the center of the Brillouin zone.

This work shows that tuning the wavelength, magnetic damping, polarization and fluence of light it is possible to find different regimes of the switching in a wide class of materials, which can possibly be even more efficient than those

demonstrated before. The examples of suitable materials are ferrites (garnet, spinel, ortho- and hexaferrites, ferric borates, and magnetite)[25], perovskites, spin glasses. For instance, a strong contribution to single ion magnetic anisotropy is typical for $3d$ (e.g., $Mn^{2+}$, $Mn^{3+}$, $Cr^{2+}$, $Cr^{3+}$, $Fe^{2+}$, $Co^{2+}$, $Co^{3+}$, $Ni^{2+}$, $Ni^{3+}$), $4d$ ($Ru^{3+}$, $Ru^{4+}$), $5d$ ($Ir^{3+}$, $Ir^{4+}$), and $4f$ ($Ce^{3+}$, $Tb^{3+}$) elements. Finding novel families of magnetic dielectrics switchable with laser pulses is crucial for future applications of photo-magnetism.

## Methods

**Materials**. The monocrystalline cobalt-doped yttrium iron garnet (YIG:Co) thin film with composition $Y_2CaFe_{3.9}Co_{0.1}GeO_{12}$ was deposited on gadolinium–gallium garnet (GGG) substrate[23]. The miscut angle of the GGG substrate has been about 4°. The dopant $Co^{2+}$ and $Co^{3+}$ ions substitute $Fe^{3+}$ in both tetrahedral and octahedral sublattices, which are coupled antiferromagnetically. The Co-ions enhance the magnetocrystalline anisotropy and result in a strong photo-magnetic effect, allowing to manipulate the magnetization with light[24]. In this sample the magnetic anisotropy constants both cubic ($K_1 = -8.4 \times 10^3$ erg cm$^{-3}$) and uniaxial ($K_U = -2.5 \times 10^3$ erg cm$^{-3}$) were measured at room temperature. Incorporation of Co-ions also increases the Gilbert damping parameter to a large value of $\alpha = 0.2$, which was measured using ferromagnetic resonance technique. Both tetrahedral and octahedral $Fe^{3+}$ ions are substituted by $Ge^{4+}$ ions to decrease the saturation magnetization ($M_S = 7$ G). $Ca^{2+}$ dopants enter the dodecahedral sites and take care of charge compensation. The Curie temperature was 455 K.

As single ion contribution to magnetic anisotropy from Co-ions is orders of magnitude higher than that of Fe ions[21], a single photon can control spins in a large garnet volume. For YIG:Co film with the thickness of 7.5 μm, the minimum pump intensity required for the switching was 34 mJ cm$^{-2}$. YIG:Co film is a ferrimagnetic dielectric transparent in the near infrared spectral range[18]. For photon energy at 0.95 eV the total absorption in the film is about 12%. It means that the density of absorbed photons is $3.6 \times 10^{19}$ cm$^{-3}$ and a single-photon switches the magnetization in 28 nm$^3$ of the garnet.

**Experimental technique for photo-magnetic imaging**. To investigate the magnetization switching induced by femtosecond linearly polarized laser pulses in YIG:Co thin film, we employed the technique of magneto-optical polarized microscopy in Faraday geometry. The magnetic contrast in polarizing microscope comes from the fact that magnetic domains with different out-of-plane magnetization component will result in different angles of the Faraday rotation. The light emitted from the LED source, passing through the polarizer will acquire different polarization rotation in different magnetic domains. The differences can be visualized with the help of an analyzer and a CCD camera. The obtained images visualize magnetic domains. The garnet film was excited with one or and multiple ($N$) pulses. The duration of each pulse was about 50 fs and pulse-to-pulse separation $t_R$ was varied. The pump beam with the fluence 250 mJ cm$^{-2}$ was focused to a spot about 130 μm in diameter. Further increasing the fluence or the number of pulses does not increase domain size considerably, but instead leads to heating of the sample. The wavelength of the pump pulses was tuned within the range between 1000 nm and 1500 nm. The images of magnetic domains were taken before and after the pump excitation. Taking the difference between the images we deduced photo-induced effects on the magnetization in the garnet. In YIG:Co thin film with small miscut angle four types of magnetic domains were observed. Any of the four domains with magnetization states 1, 4, 5, and 8 (orientations are shown in Fig. 1b) could be stabilized by applying and removing an in-plane field directed along one of the four easy-axes of the magnetization <110>. The magneto-optical contrasts for 5 (or 4) and 8 (or 1) domains at off-normal incidence of the probe are almost the same. However, 8 and 1 domains are always larger than 5 and 4 domains. The magnetization processes and domain structure analysis are systematically discussed in ref. 23 and references therein. To acquire a multi-domain state as shown in Fig. 2, the sample was brought into 8 state by applying an external magnetic field of 80 mT along the [1−10] axis. After removing the field the sample turns into a state with large 8 domains and 4 small domains. Afterwards, applying 2 mT magnetic field along the [110] axis we convert a part of large 8 domains and a part of 4 domains into large 1 and small 5 domains, respectively. This four-domain state with orthogonal in-plane magnetization components was used for demonstration of the photo-magnetic recording by scanning polarized laser beam (see Fig. 2). However, for experiments of switching in domains shown in Figs. 3−5 the demagnetization initial state with only 8 and 4 domains have been used. In these cases, the normalized switched area is a good quantity to describe the switching visualized in the magneto-optical experiment. After the magnetic field is removed, the pattern stays unchanged for at least several days due to the non-zero coercivity in YIG:Co. All measurements were done in zero applied magnetic field and at room temperature.

Figure 5 shows how the switching depends on pulse-to-pulse separation $t_R$ in the experiment. The switching was observed for single pump and multiple-pulse ($N = 5$) excitation with pulse-to-pulse separation time in the range from 1 ms

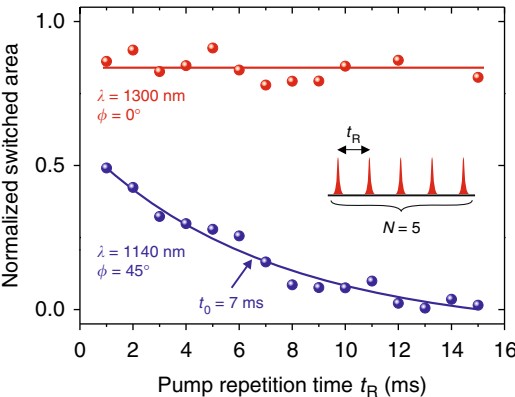

**Fig. 5** Photo-magnetic recording in YIG:Co with delay between pump pulses. The study was performed by means of analysis of normalized switched area with $N = 5$ laser pump pulses as a function of the pump pulse-to-pulse separation time $t_R$. The red solid line is a guide to the eye. The blue solid line was fitted using exponential function with the characteristic time $t_O = 7$ ms

## Data availability

The data that support the findings of this study are available from the corresponding authors on reasonable request.

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

to 20 ms. Although for pump pulses at $\lambda = 1300$ nm there is no visible accumulation effect i.e., the switched domain area does not depend on $t_R$, accumulation is clearly important for the case of $\lambda = 1140$ nm. An increase of $t_R$ results in a decrease of the switched area. It can be explained by a movement of the domain walls due to magnetostatic effects that typically occurs on the millisecond time scale[22]. With increasing fluence (see Figs. 3 and 5) the normalized value of switched area was stabilized below the maximal level limited by laser spot. We note, that the image of switched domain was performed by means of standard magneto-optical imaging at Faraday geometry about 10 ms after excitation with a single pump pulse.

**Theoretical model of photo-magnetic switching**. The expression for the photo-magnetic tensor in Voigt notation is:

$$
\hat{\beta}_V = \begin{pmatrix}
b_{1111} & b_{1122} & b_{1133} & 0 & 0 & 0 & 0 & 0 & 0 \\
b_{1122} & b_{1111} & b_{1133} & 0 & 0 & 0 & 0 & 0 & 0 \\
b_{3311} & b_{3311} & b_{3333} & 0 & 0 & 0 & 0 & 0 & 0 \\
0 & 0 & 0 & b_{2323} & 0 & 0 & b_{2332} & 0 & 0 \\
0 & 0 & 0 & 0 & b_{3131} & 0 & 0 & b_{3131} & 0 \\
0 & 0 & 0 & 0 & 0 & b_{1212} & 0 & 0 & b_{1221} \\
0 & 0 & 0 & 0 & 0 & b_{3131} & 0 & 0 \\
0 & 0 & 0 & 0 & b_{2323} & 0 & 0 & b_{2323} & 0 \\
0 & 0 & 0 & 0 & 0 & b_{1221} & 0 & 0 & b_{1212}
\end{pmatrix}, \quad (3)
$$

We take the action of the laser pulse as a perturbation to the initial state of the magnetization in the film, obtained from the LLG-equation Eq. (1). Depending on the axis of linear polarization of light it gives:

$$
\mathbf{E}||[100]([010]): \begin{cases} \theta(0) = \theta_0 \pm A \frac{1}{1+\alpha^2} \left( a \sin(2\varphi_0)\sin\theta_0 \mp \frac{\alpha}{2} b \sin(2\theta_0) \right) \\ \varphi(0) = \varphi_0 + A \frac{1}{1+\alpha^2} \left( b \cos\theta_0 \pm \alpha a \sin(2\varphi_0) \right) \end{cases}
$$

$$
\mathbf{E}||[110]([1\bar{1}0]): \begin{cases} \theta(0) = \theta_0 - \frac{1}{2} A \frac{\alpha}{1+\alpha^2} \left( b + c \sin(2\varphi_0) \right) \sin(2\theta_0) \\ \varphi(0) = \varphi_0 + A \frac{1}{1+\alpha^2} \left( b \pm c \sin(2\varphi_0) \right) \cos\theta_0 \end{cases} \quad (4)
$$

where $\varphi_0$ and $\theta_0$ denote the ground state angles of the initial domain with respect to [100] and [001] axes, correspondingly. $a = a(\lambda)$, $b = b(\lambda)$, and $c = c(\lambda)$ are the parameters, which can be expressed through photo-magnetic tensor coefficients and $A = \gamma M_s \langle E^2 \rangle \Delta t$, where $\langle E^2 \rangle$ is the average intensity of the field in the pulse and $\Delta t \approx 20$ ps is the characteristic time of the emergence of the magnetic domain taken from the experiment[10]. After time interval of the order of $\Delta t$ due to relaxation of the photo-excited Co-electrons, the magnetization dynamics can be regarded as free (see Fig. 1). As it is seen from Eq. (4), the initial perturbation strongly depends on the Gilbert damping, initial state and the intensity of light; these are parameters which are crucial for the switching. Wavelength dependence of the efficiency of switching implies wavelength dependence of the photo-magnetic tensor coefficients, namely $a(\lambda)$, $b(\lambda)$, and $c(\lambda)$.

## Acknowledgements

We acknowledge support from the project TEAM/2017-4/40 of the Foundation for Polish Science co-financed by the EU within the ERDFund, the Netherlands Organization for Scientific Research (NWO), RSF (grant No. 17-12-01333) and the programme 'Leading Scientist' of the Russian Ministry of Education and Science (14.z50.31.0034). We thank A. Maziewski and Th. Rasing for continuous support.

## Author contributions

A.S. conceived the project with contributions from A.K. and A.V.K. The measurements were performed by K.S. and A.S. The model was designed and the simulations were performed by M.D.D, K.A.Z, and A.K.Z.. A.S., A.K., and A.V.K. co-wrote the manuscript with contributions from K.S. and A.K.Z. The project was coordinated by A.S.

## Additional information

**Competing interests:** The authors declare no competing interests.

