## [Peer Review File · Nature Communications]

Reviewers' comments:

Reviewer #1 (Remarks to the Author):

Stupakiewicz et al. report on a detailed and quite interesting study of magnetization switching in iron garnets by single and multiple femtosecond laser pulses. Wavelength, (linear) polarization and pump fluences are mapped out experimentally, and the switching is reproduced by LLG modelling including an optically induced spin-torque term. Resonances are identified and selection rules derived.

Although solid work that certainly is recommended to be published in a more topical journal, the scientific impact and conceptual innovation is not considered sufficient to recommend publication in Nature Communications. In particular, many of the experimentally observed phenomena as well as the qualitative explanation for the very same material have already been published by the authors in a recent Nature article [10]. In that article, the authors already show very similar results for:

- Single pulse switching (while to me the extension to multipulse switching in combination with a scanning laser focus as shown in Figure 3 is not of significant added value; corresponding Kerr microscope images as shown in Figure 2b are less convincing)
- The pump wavelength dependence for [100] polarization with a resonance at 1200 nm
- Interpretation in terms of specific transitions reported in literature, although in the present work this is done in more detail
- Different response between [100] and [110] polarization
- The pump fluence dependence at 1300 nm, with a threshold of 34 mJ/cm^2 .
- The interpretation in terms of response tensors, as discussed in the Methods section of [10], although a slightly different formulation seems to have been chosen in the present work.

In passing, it is not appreciated that the manuscript does not state which results were already reported in [10], and which results are conceptually really new.

New with respect to [10] are:

- The polarization dependence depicted in Figure 4b, although there could well be some link to the data in the Extended Data Figure 3 in [10].
- Data on the resonance at 1140 nm.
- The LLG modelling of the switching event, resulting in the Phase diagrams represented in Fig. 1 c,d.

Disclaimer: I may have overlooked some other subtle differences, but I would consider it the task of the authors to shed light on this in their manuscript, and not the task of the reader having to search for it.

In particular the LLG simulations, but also the new experimental data, are interesting and are recommended to be published in a more topical journal. Before doing so, the manuscript should be made more explicit as to which aspects of the paper are new, and which were already reported before.

Reviewer #2 (Remarks to the Author):

The work by Stupakiewicz et al. uses optically-induced spin-torque in a ferromagnetic insulator to manipulate the state of the magnetization. A very careful study that adds more understanding and deep insights to this intriguing and complex question, how ultrashort light pulses can be used to manipulate the magnetization. Here they discuss a detailed way the selection rules of the excitation process, namely the role of crystal axis and light polarization that goes into the optical transitions. This clearly shows the role of the induced magnetization by the light, which is

connected to these transitions. It is a very good and excellent work, with careful experiments and well written and presented, and opens up novel avenues for photo-magnetism. I fully support a publication in Nature Communications without any restraint.

Reviewer #3 (Remarks to the Author):

The paper by A. Stupakiewicz et al. entitled "Selection Rules for All-Optical Magnetic Recording in Iron Garnet" presents optical-induced polarization- and wavelength-dependent magnetization switching in Co-doped iron garnets and discusses the key features of this phenomenon and the underlying switching mechanism. The topic is very important for the field of femtomagnetism and the manuscript considers fundamental issues using the example of relatively simple and well-known model system. Thus, the paper could be quite attractive for a broad audience. On the other hand, the presented work is in fact an extension of recently published Ref. [10] where the single-pulse switching of the magnetization in the same (or very similar) samples has been already demonstrated and the key points of the underlying mechanism were explained. Moreover, Ref. [10] clearly shows the single-pulse character of the switching while the present paper mostly deals with a multi-pulse switching (there is only one point in Fig. 3, which shows the switching of 10% of the laser spot area by a single pulse at the highest used fluence).

Comparing to Ref. [10], the new information here contains the following points: (i) In addition to the already shown switching by the pulse polarized along [100] direction, the (less-efficient) switching by the [110] polarization is demonstrated; (ii) it is shown that the optimal photon energies for the two above cases are different; (iii) based on that, the involved electronic transitions are identified, which occurs in Co ions located at different crystallographic sites; (iv) the phenomenological model of the switching is discussed in more details and supported by simulations. Although these findings without doubts represent a significant progress in the studies, it is difficult to realize that any qualitatively new information/conclusion has been obtained/derived. On the other hand, owing to the chosen paper format, the presentation and discussion are very compressed and not easy for understanding. It is especially difficult to get the point without reading carefully the Ref. [10] and in this sense the considered manuscript really looks like a follow-up paper presenting more results on qualitatively the same issues. Thus, I am not sure that the choice of format and journal is optimal in this case. Probably, a longer paper systematically presenting the picture in more details could be much more helpful and suitable here although I completely understand the wish of authors to increase the visibility of their interesting and important results.

Regarding the presented manuscript, I have an impression that its clarity and accessibility for a general reader still can (and must) be improved and give the related questions and comments below.

1) Line 30: "... with light polarized along [100] or [010] axis..." Finally, it is clear that the [001] direction is excluded because in the experiment this is the normal to surface. However, in this place, the authors discuss the bulk properties of a cubic material and such statement can be a bit confusing.

2) Line 64: "Fixing the damping at $\alpha=0.28$, which is very close to the experimentally observed value [10,13]..." The experimentally observed value in [10] was 0.2 as well as in the present work as it follows from Methods section, line 255 (eventually, the same sample has been used in both works, [10] and this one since not only the damping but also the given measured anisotropy constants are identical in the two cases). Thus, the damping used in calculations differs from the experimental one by 40% which is not "very close", especially in the light of Fig. 1 c and d: for $\alpha=0.28$ the magnetization switches from state 1 to state 8 at a fluence exceeding certain threshold while for $\alpha=0.2$ with increasing fluence it will first switch from 1 to 5 and only at higher fluencies from 1 to 8. I would appreciate some comments on that. Moreover, the caption of

Fig. 1 again states explicitly that “the calculations are performed for $\alpha=0.28$ ” (line 84). For this value, according to the phase diagrams c and d, only the switching 1- \rightarrow 8 is possible. This contradicts panels e and f where the trajectories 1- \rightarrow 5 requiring smaller damping (0.2 suits) and 1- \rightarrow 4 requiring the damping even smaller than 0.2, are shown. I would ask for a consistency here.

3) In lines 67-69 and 270 “in-plane” and “perpendicular” components are mentioned while the plane itself is not defined. Eventually, this is the plane of sample surface (001) but it would be better if the authors say this.

4) Lines 100-102: “Both polarizations are different from those used for the switching in ref. 10 indicating that the magnetic recording relies on a different microscopic mechanism which has not been reported before.” I am quite not sure that it is suitable to talk about a different microscopic mechanism here. According to my understanding, the mechanism is exactly the same, described by Eqs. (1) and (2). The fact that different electronic transitions at different ions are involved is rather a detail than a principal difference.

5) With regard to Fig. 2, the authors discuss the switching from state 1 to state 8 (see line 106). I can accept that the authors know the direction of magnetization in the initial state, which can be determined by the procedure of initial state preparation with the help of external magnetic field applied in a special way. Unfortunately, the authors do not describe this procedure (in contrast to Ref. [10]), which I believe can be included in the Method section in the next version. How does the final state associated with the state 8 but not 5? As far as I understand, these two states are indistinguishable in the Faraday geometry at the normal incidence. However, they can be distinguished by using the off-normal incidence and rotating the sample. Was such an effort applied in the experiment? This could also answer the question of actual damping in the scale of phase diagrams Fig. 1 c, d (cf. point 2 above) In any case I think that the description of experiment should provide more details with respect to that.

6) The caption of Fig. 2 gives the laser fluence and the repetition rate. I believe it would be worth to give also the spot size (although it can be found in Methods) and the scan velocity. It would be even better to give directly the effective number of pump pulses per point of image.

7) Fig. 3 shows the “normalized switched area” which I think has not been introduced in the text although it seems that the inset has been prepared for that (it is introduced in Ref. [10] however the measure of the spot size is not mentioned, which can be $1/e$ or $1/e^2$ or whatever).

8) Lines 113-116: “Note that the appearance of a domain after a certain number of pulses is a very reproducible effect that is determined by a balance between the repetition rate of the pulses and magnetostatic field driven domain wall motion (for more details see Methods and Extended Data Fig. 1).” However, in Methods, one can find only the following phrase related to this issue (lines 287-289): “An increase of t_R results in a decrease of the switched area. It can be explained by a movement of the domain walls due to magnetostatic effects that typically occurs on the millisecond time scale [21].” I would not say that this clarifies the role of domain wall motion in the observed switching and would expect to find indeed more extended discussion here. Is this the case that the optically-induced spin-torque given by Eq. (2) represents only the first, ultrafast stage of the switching mechanism, while the second, slow one is based on the movement of the domain walls? If the switched domains are formed on the millisecond time scale then can one really talk about an ultrafast recording in this case even if a single pump pulse has been applied? It seems that the authors should provide more detailed, precise, and clear description of the whole switching mechanism and specify what is the “ultrafast recording” in this context and how fast it is in the reality.

9) There are three dependencies shown in Fig. 3 measured for different fluencies. With increasing fluence they stop at 20, 10, and 5 pump pulses, respectively, but there is no clear saturation in all of them, especially in that for the highest fluence. Will the switched area increase with further

increasing of the number of pump pulses? Similar question is related to Fig. 4 c where the dependence for 1300 nm stops at half of the fluence range used for 1140 nm: what will happen to the switched area if the fluence is increased further? Interestingly, for all shown dependencies in this paper as well as in Ref. [10], the measured normalized switched area does not exceed 0.5. What is the origin of this magic number?

10) Caption to Fig. 4 b, lines 137, 138: "Solid lines are fit by $\cos(2\phi)$ function". Indeed, this function is contained in Eq. (4). However, solid lines in Fig. 4 b have clear character of $\cos(\phi)^2$ function (which is of course $\cos(2\phi) + \text{const}$, where const is very close to 1). Since a coincidence is highly unlikely here, are there some corresponding restrictions on the parameters in Eq. (4)?

11) Line 143: remove "nm".

12) Section Methods, Materials: I am missing an exact composition of samples like it has been given in Ref. [10].

13) Line 304: "where ϕ_0, θ_0 denote the ground state angles of the initial domain". I think, these angles should be introduced with respect to crystallographic directions.

REPLY TO REFEREES'

Referee 1

I may have overlooked some other subtle differences, but I would consider it the task of the authors to shed light on this in their manuscript, and not the task of the reader having to search for it. In particular the LLG simulations, but also the new experimental data, are interesting and are recommended to be published in a more topical journal. Before doing so, the manuscript should be made more explicit as to which aspects of the paper are new, and which were already reported before.

Reply to Referee 1:

We thank the Referee for the criticism and we would like to stress that:

- our paper is the very first report which reveals the possibility of ultrafast all-optical magnetic recording with two different pairs of polarizations of light. Depending on the pump wavelength, different pairs must be used to record and restore a magnetic domain. Such a possibility has not been either predicted or anticipated before.
- our paper reveals new mechanism of light-matter interaction which facilitates all optical magnetic recording in dielectrics. The knowledge of the mechanisms, i.e. electronic transitions, responsible for the effect is the main obstacle preventing any quantitative modeling of light-matter interactions. Therefore, we see our paper as a breakthrough that opens up the way for theoretical and computational studies of all-optical magnetic recording and ultrafast photo-magnetic phase-transitions in dielectrics.
- the model reported in the present paper predicts the possibility of magnetic recording with two different pairs of light polarizations as well as reveals the phase diagram for switching together with the trajectories of the laser-induced magnetization dynamics. None of these theoretical results was possible without a substantial upgrade of the model from Ref. [10].

Nevertheless, we admit that the previous version of the manuscript was lacking explicit statements showing which aspects of the paper are really new. To meet the criticism we have changed the abstract and introduction of the paper.

The abstract and text have been changed accordingly (see issues 1-2 in the summary of changes).

Referee 3

Thus, the paper could be quite attractive for a broad audience. On the other hand, the presented work is in fact an extension of recently published Ref. [10] where the single-pulse switching of the magnetization in the same (or very similar) samples has been already demonstrated and the key points of the underlying mechanism were explained. Moreover, Ref. [10] clearly shows the single-pulse character of the switching while the present paper mostly

deals with a multi-pulse switching (there is only one point in Fig. 3, which shows the switching of 10% of the laser spot area by a single pulse at the highest used fluence).

Comparing to Ref. [10], the new information here contains the following points: (i) In addition to the already shown switching by the pulse polarized along [100] direction, the (less-efficient) switching by the [110] polarization is demonstrated; (ii) it is shown that the optimal photon energies for the two above cases are different; (iii) based on that, the involved electronic transitions are identified, which occurs in Co ions located at different crystallographic sites; (iv) the phenomenological model of the switching is discussed in more details and supported by simulations. Although these findings without doubts represent a significant progress in the studies, it is difficult to realize that any qualitatively new information/conclusion has been obtained/derived.

Reply to Referee 3:

We are happy that the Referee finds our paper attractive and absolutely correctly summarizes the main achievements of our work. Nevertheless, we would like to emphasize that the possibility of magnetic recording with different pairs of polarizations of light has not been either predicted or even anticipated before. This qualitatively new finding is now stressed in the abstract and introduction of the paper.

The abstract and text have been changed accordingly (see issues 1-2 in the summary of changes).

Referee 3.1

Regarding the presented manuscript, I have an impression that its clarity and accessibility for a general reader still can (and must) be improved and give the related questions and comments below.

1) Line 30: "... with light polarized along [100] or [010] axis..." Finally, it is clear that the [001] direction is excluded because in the experiment this is the normal to surface. However, in this place, the authors discuss the bulk properties of a cubic material and such statement can be a bit confusing.

Reply to Referee 3.1:

To meet the criticism we have added additional sentence which explains why we do not consider [001] in the model (see issues 3 and 4 in the summary of changes).

Referee 3.2

2). Line 64: "Fixing the damping at $\alpha=0.28$, which is very close to the experimentally observed value [10,13]..." The experimentally observed value in [10] was 0.2 as well as in the present work as it follows from Methods section, line 255 (eventually, the same sample has been used in both works, [10] and this one since not only the damping but also the given measured anisotropy constants are identical in the two cases). Thus, the damping used in calculations differs from the experimental one by 40% which is not "very close", especially in the light of Fig. 1 c and d: for $\alpha=0.28$ the magnetization switches from state 1 to state 8

at a fluence exceeding certain threshold while for $\alpha=0.2$ with increasing fluence it will first switch from 1 to 5 and only at higher fluencies from 1 to 8. I would appreciate some comments on that. Moreover, the caption of Fig. 1 again states explicitly that “the calculations are performed for $\alpha=0.28$ ” (line 84). For this value, according the phase diagrams c and d, only the switching 1->8 is possible. This contradicts panels e and f where the trajectories 1->5 requiring smaller damping (0.2 suits) and 1->4 requiring the damping even smaller than 0.2, are shown. I would ask for a consistency here.

Reply to Referee 3.2: We are thankful to the Referee who has carefully read the paper and helped us to reveal a number of confusing misprints. We would like to note that the value of the Gilbert damping $\alpha=0.2$, given in Methods was measured using the technique of ferromagnetic resonance (we added this information in Methods) and indeed differs from the value which we assumed in theory. However, it is known that the Gilbert damping constant deduced from FMR experiments, where the amplitude of magnetization oscillations is small, can be smaller than the effective damping seen in large-amplitude processes [ref. 14 - A. P. Malozemoff, and J. C. Slonczewski, Magnetic Domain Walls in Bubble Materials: Advances in Materials and Device Research. Vol. 1. Academic press, 2016]. As a matter of fact, the best agreement between the experimentally observed and simulated dynamic is obtained for $\alpha=0.28$. Moreover, we also would like to note that precise experimental determination of the damping constant of a heavily damped oscillator is a rather challenging task. Therefore we think damping constants equal to 0.28 and 0.2 are rather close. Nevertheless, we understand the point of the Referee.

The text and captions have been corrected accordingly (see issues 6 and 17 in the summary of changes).

Referee 3.3

3) In lines 67-69 and 270 “in-plane” and “perpendicular” components are mentioned while the plane itself is not defined. Eventually, this is the plane of sample surface (001) but it would be better if the authors say this.

Reply to Referee 3.3: The sample plane is defined in text (see our reply to Referee 3.1 and issues 3 and 5).

Referee 3.4

4) Lines 100-102: “Both polarizations are different from those used for the switching in ref. 10 indicating that the magnetic recording relies on a different microscopic mechanism which has not been reported before.” I am quite not sure that it is suitable to talk about a different microscopic mechanism here. According to my understanding, the mechanism is exactly the same, described by Eqs. (1) and (2). The fact that different electronic transitions at different ions are involved is rather a detail than a principal difference.

Reply to Referee 3.4:

Although we can partly agree with the Referee and admit that expressions Eqs. (1) and (2) derived from the laws of thermodynamics are still valid for both [100] and [110] polarizations of light, we are convinced that it is quite suitable to talk about new mechanism of light-matter interaction which facilitates all-optical magnetic recording. Our work shows that the photo-magnetic anisotropy, which is currently treated phenomenologically, is due to electronic transitions in Co-ions in both tetrahedral and octahedral environments. It makes a principle difference in the field and, in particular, opens up the way for theoretical and computational studies of all-optical magnetic recording in magnetic dielectrics.

The text have been corrected accordingly (see issues 7 and 10 in the summary of changes).

Referee 3.5

5) With regard to Fig. 2, the authors discuss the switching from state 1 to state 8 (see line 106). I can accept that the authors know the direction of magnetization in the initial state, which can be determined by the procedure of initial state preparation with the help of external magnetic field applied in a special way. Unfortunately, the authors do not describe this procedure (in contrast to Ref. [10]), which I believe can be included in the Method section in the next version. How does the final state associated with the state 8 but not 5? As far as I understand, these two states are indistinguishable in the Faraday geometry at the normal incidence. However, they can be distinguished by using the off-normal incidence and rotating the sample. Was such an effort applied in the experiment? This could also answer the question of actual damping in the scale of phase diagrams Fig. 1 c, d (cf. point 2 above) In any case I think that the description of experiment should provide more details with respect to that.

Reply to Referee 3.5: The type of the domain structure and magnetization orientations at the domains in YIG:Co thin films have been identified and reported in many previous papers (Ref.[22] and Refs. [21,23]) in detail.

We have added in Methods (see issues 8,9,18 in the summary of changes).

Referee 3.6

6) The caption of Fig. 2 gives the laser fluence and the repetition rate. I believe it would worth to give also the spot size (although it can be found in Methods) and the scan velocity. It would be even better to give directly the effective number of pump pulses per point of image.

Reply to Referee 3.6: We have added in caption of Figure 2 (see issue 11 in the summary of changes).

Referee 3.7

7) Fig. 3 shows the “normalized switched area” which I think has not been introduced in the text although it seems that the inset has been prepared for that (it is introduced in Ref. [10] however the measure of the spot size is not mentioned, which can be $1/e$ or $1/e^2$ or whatever).

Reply to Referee 3.7: We have added in the caption of Fig. 3 (see issue 14 in the summary of changes).

Referee 3.8

8) Lines 113-116: “Note that the appearance of a domain after a certain number of pulses is a very reproducible effect that is determined by a balance between the repetition rate of the pulses and magnetostatic field driven domain wall motion (for more details see Methods and Extended Data Fig. 1).” However, in Methods, one can find only the following phrase related to this issue (lines 287-289): “An increase of t_R results in a decrease of the switched area. It can be explained by a movement of the domain walls due to magnetostatic effects that typically occurs on the millisecond time scale [21].” I would not say that this clarifies the role of domain wall motion in the observed switching and would expect to find indeed more extended discussion here. Is this the case that the optically-induced spin-torque given by Eq. (2) represents only the first, ultrafast stage of the switching mechanism, while the second, slow one is based on the movement of the domain walls? If the switched domains are formed on the millisecond time scale then can one really talk about an ultrafast recording in this case even if a single pump pulse has been applied? It seems that the authors should provide more detailed, precise, and clear description of the whole switching mechanism and specify what is the “ultrafast recording” in this context and how fast it is in the reality.

Reply to Referee 3.8: Thank you for bringing up this point. The switched domains are observed on the millisecond time scale using classical magneto-optical microscopy. In this case the images were acquired about 10 ms after excitation with a single pump pulse. However these domains are formed really at ultrafast time scale after single pump pulse, that we demonstrated in previous paper Ref.[10] using unique time-resolved single-shot imaging.

The text in Methods has been changed accordingly (see issue 12 in the summary of changes):

Referee 3.9

9) There are three dependencies shown in Fig. 3 measured for different fluencies. With increasing fluence they stop at 20, 10, and 5 pump pulses, respectively, but there is no clear saturation in all of them, especially in that for the highest fluence. Will the switched area increase with further increasing of the number of pump pulses? Similar question is related to Fig. 4 c where the dependence for 1300 nm stops at half of the fluence range used for 1140 nm: what will happen to the switched area if the fluence is increased further? Interestingly, for all shown dependencies in this paper as well as in Ref. [10], the measured normalized switched area does not exceed 0.5. What is the origin of this magic number?

Reply to Referee 3.9: See our reply to 3.8

We demonstrated the quasi-static imaging using classical magneto-optical Faraday scheme. In this case the images were acquired about 10 ms after excitation with a single pump pulse. Thus we observed the remnant domains after long time in comparison to the switching time of about 20 ps.

The text has been changed accordingly (see issues 13 and 19 in the summary of changes):

Referee 3.10

10) Caption to Fig. 4 b, lines 137, 138: “Solid lines are fit by $\cos(2\phi)$ function”. Indeed, this function is contained in Eq. (4). However, solid lines in Fig. 4 b have clear character of $\cos(\phi)^2$ function (which is of course $\cos(2\phi)+const$, where $const$ is very close to 1). Since a coincidence is highly unlikely here, are there some corresponding restrictions on the parameters in Eq. (4)?

Reply to Referee 3.10: In this figure the solid lines are fitted by $\cos(2\phi)$ and $\cos(2\phi-\pi/4)$ functions. The $\cos(2\phi)$ -like dependence is typical for the photo-magnetic effect.

We have added in the caption of Fig. 4 (see issue 15 in the summary of changes).

Referee 3.11

11) Line 143: remove “nm”.

Reply to Referee 3.11: This flaw was removed.

Referee 3.12

12) Section Methods, Materials: I am missing an exact composition of samples like it has been given in Ref. [10].

Reply to Referee 3.12: We have added the sample composition in the Method Section (see issue 16 in the summary of changes).

Referee 3.13

13) Line 304: “where ϕ_0 , θ_0 denote the ground state angles of the initial domain”. I think, these angles should be introduced with respect to crystallographic directions.

Reply to Referee 3.13: The manuscript has been changed accordingly (see issue 20 in the summary of changes).

SUMMARY OF CHANGES:

Issue	Line	Change
1	11	We have added in abstract: “All-optical magnetic recording with femtosecond laser pulses triggered intense discussions about microscopic mechanisms responsible for this phenomenon.”
	16	“The all-optical magnetic switching under both single pulse and multiple-pulse sequences can be achieved at room temperature, in narrow spectral ranges with two pairs of light polarization, either along [110] and [1-10] or [100] and [010] crystallographic axes of the garnet.”
2	26	We have added in text of introduction: “Ultrafast light-matter interactions in the case of metals are often explained with the help of a simplistic temperature model, where features of the electronic structure hardly play any role [9].”
	40	“In this paper, we demonstrate that contrary to previous reports [10], all-optical magnetic recording can be achieved not with one, but with two pairs of light linear polarizations. Depending on the pump wavelength, different pairs of polarizations must be used to record and erase a magnetic domain. Such a possibility has not been either predicted or anticipated before. Note that the knowledge of the mechanisms, i.e. electronic transitions, responsible for the effect is the main obstacle preventing any quantitative modeling of light-matter interactions. Here, we reveal these transitions and thus open up the way for theoretical and computational studies of all-optical magnetic recording and ultrafast photo-magnetic phase-transitions in dielectrics.”
3	31	Few words have been added and the sentence now states: “Recently it has indeed been shown that using a single 50 fs laser pulse one can permanently switch the magnetization in Co-doped yttrium iron garnet thin film (YIG:Co) with (001)-plane of the sample [10]”
4	62	Few words have been added and the sentence now states: “The number of independent tensor components can be found taking into account the $4mm$ point group for (001)-oriented YIG:Co film [11,12] (see Fig. 1a) and the fact that the tensor β_{ijkl} must be invariant with respect to permutations of the last two indices.”
5	81	Few words have been added and the sentence now states: “This model demonstrates different trajectories of the switching (see Fig. 1e) between: (i) only in-plane magnetization components (between [110] and [1-10] axes, see “1-4” trajectory); (ii) only perpendicular magnetization components (between [001] and [00-1] axes, see “1-5” trajectory); (iii) simultaneously in-plane and perpendicular magnetization components (between [111] and [1-1-1] axes, see “1-8” trajectory).”
6	86	We have added in text: “The Gilbert damping constant was treated as a

		fitting parameter in the simulations. The best agreement of our simulations with the experimental data reported here was achieved for $\alpha=0.28$. Note that the previously mentioned experimental value of $\alpha=0.2$ [10,13] was measured using the technique of ferromagnetic resonance, where the amplitudes of magnetization oscillations are small. It is well known, however, that the damping increases with the amplitude of the magnetization oscillations [14].”
7	92	Few words have been added and the sentence now states: “Hence the simulations suggest that in addition to the photo-magnetic recording reported in ref. 10, the symmetry of the garnet films supports the possibility of the magnetic switching using other light polarizations than [100] and [010], for which the mechanism of the switching must rely on excitation of other electronic transitions with other selection rules.”
8	105	We have added in caption of Fig. 1: “The trajectories calculations are performed for various damping and fluence, based on the phase diagrams to help illustrate the switching. Trajectories of the switching from state “1” to “8” was performed for $\alpha=0.28$. The switching paths from “1” to “4” and from “1” to “5” were simulated for $\alpha=0.12$.”
9	121	We have added in text: “The domain structure in YIG:Co thin films has been studied and reported in detail in refs. 21-23 together with a description of the experimental procedure of defining the magnetization in the domains.”
10	128	We have added in text: “Both polarizations are different from those used for the switching in ref. 10 indicating that here it most likely proceeds along different microscopic mechanism, i.e. relies on excitation of different electronic transitions than those reported earlier.”
11	135	We have added in caption of Fig. 2: “The pump beam was focused to a spot with the radius $r=65 \mu\text{m}$. The scan velocity was $200 \mu\text{m s}^{-1}$ with 1000 pulses per second.”
12	140	Few words have been added and the sentence now states: “Figure 3 shows how the switched area increases with an increase of number of pump pulses N using quasi-static magneto-optical imaging in Faraday geometry within 10 ms frame after excitation with a single pump pulse (see Methods) (see Fig. 3a).”
13	146	We have added in text: “The process of domain formation using multiple pulses for 105 mJ cm^{-2} is following: after first pump pulse the magnetization was switched, however the switched area was not stable enough. The next pulses resulted in an expansion and thus stabilization of the recorded domain. As a matter of fact, the switched domains have been clearly and reproducibly seen after $N=5$ (see. Fig. 3).”
14	155	We have added in caption of Fig. 3: “The normalized switched area was calculated as the ratio of the recorded domain area (the black large domain limited by blue circle) to the area of pump laser spot πr^2 (limited by red

		circle).”
15	174	Few words have been added and the sentence now states: “Solid lines are a fit by $\cos(2\phi)$ -like function which is the typical for the photo-magnetic effect [10,21].”
16	205	We have added in Methods: “The monocrystalline cobalt-doped yttrium iron garnet (YIG:Co) thin film with composition $\text{Y}_2\text{CaFe}_{3.9}\text{Co}_{0.1}\text{GeO}_{12}$ was deposited on gadolinium-gallium garnet (GGG) substrate [22].”
17	212	We have added in Methods: “Incorporation of Co ions also increases the Gilbert damping parameter to a large value of $\alpha=0.2$, which was measured using ferromagnetic resonance technique.”
18	238	We have added in Methods: “Main four magnetization states (5, 6, 7 and 8) are obtained as follows. First, the sample is brought into "8" state by an external magnetic field of $\mu_0H = 80$ mT applied along the [1-10] direction. Second, the field is removed and the sample turns into a state with "8" (large domains) and "4" (small domains) domains. Third, a magnetic field $\mu_0H = 2$ mT for a short time applied along the [110] direction favors "1" large domains (simultaneously "5" small domains) and results in the final pattern. After the magnetic field is removed, the pattern stays unchanged for at least several days due to the non-zero coercivity in YIG:Co.”
19	254	We have added in Methods: “With increasing fluence (see Fig. 3 and Supplementary Figure 1) the normalized value of switched area was stabilized below the maximal level limited by laser spot. We note, that the image of switched domain was performed by means of standard magneto-optical imaging at Faraday geometry about 10 ms after excitation with a single pump pulse.
20	276	Few words have been added and the sentence now states: “where φ_0 and θ_0 denote the ground state angles of the initial domain with respect to [100] and [001] axes, correspondingly.”
21	326	We have added in References: [14] Malozemoff, A. P., and Slonczewski, J. C., Magnetic Domain Walls in Bubble Materials: Advances in Materials and Device Research . Vol. 1. (Academic press, 2016).

Reviewers' comments:

Reviewer #2 (Remarks to the Author):

Very nice work and everything satisfyingly answered. I suggest to accept the manuscript as is for Nature Communications.

Reviewer #3 (Remarks to the Author):

The authors considerably improved the manuscript in its second version better emphasizing the achievements of present work compared to that of the previous one. The last paragraph of the introduction (issue 2 in the summary of changes) looks almost convincing. However, here, I am missing a clear explanation of the importance of the excitation by two pairs of pump polarizations at different energies, which is discussed as the main achievement of the work. If the authors complement their general statements with some examples or at least ideas of how and where this can be used, the doubts in the novelty of presented work will be strongly reduced.

Then, there are some points raised in my previous report where I am still waiting for reasonable answers and comments:

1) I am not satisfied by the answer to the point 5 of my previous report, more exactly by the absence of an answer to the central question of this point: how the FINAL magnetization state has been identified? (The given answer concerns only the initial state as far as I can understand.) Are there clear EXPERIMENTAL EVIDENCES of the switching to state 8 but not 5? The authors claim finally that they have an uncertainty in the damping value since the one measured with FMR may be too low for the (large-amplitude) process they focus on (see the answer to the point 2). Then a different experimental geometry (off-normal incidence of the probe) might help to identify the final state and probably verify the calculated phase diagrams or/and shed light to the actual damping value. To my point of view, this is an important question and I would appreciate an extended answer to that. Have such or similar experiments been done? If yes, what was the result? If not, then why? Are there some principal problems with an experimental identification of the final state? If it difficult or impossible to define the other (than the normal-to-surface one) projections of the final magnetization in the experiment, a corresponding discussion is expected in Methods. If it is possible then such experiment should be performed to support the discussion and conclusions given in the considered manuscript.

2) Also, the questions in point 9 of my previous report have not been answered. These questions were not about the switching mechanism (this issue has been addressed in point 8) but about reasons for limited ranges in Figs. 3 and 4.

3) Finally, I have not got the reply to my point 10 regarding $\cos(\phi)^2$ vs. $\cos(2\phi)$. Indeed, the fit in Ref. [10] to $\cos(2\phi)$ described the precession amplitude which was changing the sign (according to the chosen notations assuming the fixed phase). In contrast, the "normalized switching area" shown in Fig. 4 b is by definition a positive quantity and thus cannot be fit to $\cos(2\phi)$. Unfortunately, it is difficult to get an idea about the character of the dependence from the figure since it is plotted in polar coordinates which completely hide the shape of minima if the value is close to zero. I would ask to clarify this issue and strongly recommend to use Cartesian coordinates unless there is a very special case requiring (and allowing) the use of polar ones.

I would like to ask the authors for carefully addressing these remaining points (in particular, the point 1 above) before the final decision on the manuscript is made.

REPLY TO REFEREE #3

The authors considerably improved the manuscript in its second version better emphasizing the achievements of present work compared to that of the previous one. The last paragraph of the introduction (issue 2 in the summary of changes) looks almost convincing. However, here, I am missing a clear explanation of the importance of the excitation by two pairs of pump polarizations at different energies, which is discussed as the main achievement of the work. If the authors complement their general statements with some examples or at least ideas of how and where this can be used, the doubts in the novelty of presented work will be strongly reduced.

Reply to Referee 3:

In response to the criticism we have given specific examples of potential impact of the discovered multi-wavelength recording. In particular, we now write in “Discussion” section (lines 193-212):

“Apart from purely fundamental interest and impact on further development of theory of photo-magnetism, the discovered multi-wavelength magnetic recording in iron garnet may spur applications of photo-magnetism in information processing technology. Firstly, the possibility of recording at several wavelength shows that the wavelength for the most efficient photo-magnetic recording is not fixed but can be tuned in the range relevant to optical telecommunication systems. Secondly, our work also shows that information transferred from light to spins can be encoded not only in polarization state, but also in the wavelength and the intensity of the recording beams. In fact, our discovery shows that due to the additional degree of freedom one can increase the intensity of data traffic from light to spins. For instance, different spectral components of light can carry information which due to dispersion will be focused and recorded on different domains of a storage medium. Thirdly, the discovered selection rules allowed us to reveal the origin of the photo-magnetic centers and thus to perform quantitative estimates of the efficiency of photo-magnetic recording. In the studied films only one of forty Fe ions is substituted by Co [22-24]. It means that single-photon excitation of Co ion must be, in principle, sufficient to control the spins in a rather large volume of 28 nm³ (see Methods). Fourthly, the selection rules reported in this paper reveal the possibility of an independent control of spins coupled to octahedral and tetrahedral photo-magnetic centers. Employing the discovered degrees of freedom, polarization pulse shaping and the principles of coherent control, one can anticipate new opportunities for ultrafast optical manipulation of coherent spin waves far from the center of the Brillouin zone.”

Then, there are some points raised in my previous report where I am still waiting for reasonable answers and comments:

Referee 3.1

1) I am not satisfied by the answer to the point 5 of my previous report, more exactly by the absence of an answer to the central question of this point: how the FINAL magnetization state has been identified? (The given answer concerns only the initial state as far as I can understand.) Are there clear EXPERIMENTAL EVIDENCES of the switching to state 8 but not 5? The authors claim finally that they have an uncertainty in the damping value since the one measured with FMR may be too low for the (large-amplitude) process they focus on (see the answer to the point 2). Then a different experimental geometry (off-normal incidence of the probe) might help to identify the final state and probably verify the calculated phase diagrams or/and shed light to the actual damping value. To my point of view, this is an

important question and I would appreciate an extended answer to that. Have such or similar experiments been done? If yes, what was the result? If not, then why? Are there some principal problems with an experimental identification of the final state? If it difficult or impossible to define the other (than the normal-to-surface one) projections of the final magnetization in the experiment, a corresponding discussion is expected in Methods. If it is possible then such experiment should be performed to support the discussion and conclusions given in the considered manuscript.

Reply to Referee 3.1: The procedure allowing us to determine the final state is practically the same and based on application of an external in-plane magnetic field. We admit, however, that the procedure is destructive with respect to the domain pattern. Therefore, we did not have a possibility to do it in every experiment and must refer to previous results. Although domains with the magnetization in states “8” and “5” result in the same Faraday rotation, choosing a sample with a miscut we break the degeneracy between “5” and “8” domains. As a result, “8”-domain is always larger than “5” domain (see sketch below). It can be shown by application of an external in-plane magnetic field in different directions. For this particular sample it was done in Ref. 21. In this work we repeated the measurements and confirmed the conclusions of Ref. 21. We have been performed experiments with non-perpendicular incidence to the sample plane of the probe beam. However the obtained contrast of the images have been practically identically.

In response to the request of the Referee in the present version of the paper we answer the asked questions in “Methods” section. In particular, we now write (lines 258-267):

“For a perfectly cut (001) garnet film the magnetizations in “5” and “8” states have equal projections on the normal to the sample and thus at normal incidence the areas magnetized in states “5” and “8” look identical in polarization microscope. The magneto-optical contrasts for “5” and “8” domains at off-normal incidence of the probe are almost the same. Although in this geometry magneto-optical detection becomes sensitive to in-plane components of the magnetization, the normal component still dominates the signal so that “5” and “8” domains cannot be distinguished. Nevertheless, choosing a sample with 4° miscut angle we break the degeneracy between “5” and “8” domains [10,21]. As a result, “8” - domain is always larger than “5” domain. It can be shown by application of an external magnetic field in different directions [21].”

Referee 3.2

2) Also, the questions in point 9 of my previous report have not been answered. These questions were not about the switching mechanism (this issue has been addressed in point 8) but about reasons for limited ranges in Figs. 3 and 4.

Reply to Referee 3.2: The switched area will be almost the same after increase with further increasing of the number of pump pulses. Increasing both fluence and pulses number lead to increasing the temperature in the sample. Therefore, the combination fluence and pulse number in our experiments was limited.

We have added in “Methods” section (lines 252-254): **“Further increasing the fluence or the number of pulses does not increase domain size considerably, but instead leads to heating of the sample.”**

Referee 3.3

3) Finally, I have not got the reply to my point 10 regarding $\cos(\phi)^2$ vs. $\cos(2\phi)$. Indeed, the fit in Ref. [10] to $\cos(2\phi)$ described the precession amplitude which was changing the sign (according to the chosen notations assuming the fixed phase). In contrast, the “normalized switching area” shown in Fig. 4 b is by definition a positive quantity and thus cannot be fit to $\cos(2\phi)$. Unfortunately, it is difficult to get an idea about the character of the dependence from the figure since it is plotted in polar coordinates which completely hide the shape of minima if the value is close to zero. I would ask to clarify this issue and strongly recommend to use Cartesian coordinates unless there is a very special case requiring (and allowing) the use of polar ones.

I would like to ask the authors for carefully addressing these remaining points (in particular, the point 1 above) before the final decision on the manuscript is made.

Reply to Referee 3.3: We thank the Referee for this remark and admit that the discussion of the polarization dependence was not clear enough. We have realized that “Normalized switching area”, used for quantification of the switching efficiency, does not account for the color of the switched domain. Hence it can adequately reflect the polarization dependence only if the polarization is rotated for less than 90-degrees so that the color of the switched domain does not change.

To describe the full polarization dependence over the range of 360-degrees, one would need to introduce a new quantity - “Normalized area of relative switching”- which is equal to “Normalized switching area” taken with the sign of the z-component of the magnetization M_z .

In response to the Referee here we give the polarization dependence in Cartesian coordinates (see below) and note that the “Normalized area of relative switching” by [100]-polarization (red points) follows $\cos^2\phi$ -function very well.

We are convinced, however, that polar coordinates are the most suitable for the goals targeted by Fig.4b. In particular, these coordinates allow one to reveal the symmetry of the observed polarization dependence. This is the main reason why in the paper we prefer to give the polarization dependence in polar coordinates. In order to avoid misunderstandings the figure 4b is simplified by removing fits and changing on "guide to the eye".

We have replaced in caption of Figure 4b (line 174) “Solid lines are a fit by $\cos(2\phi)$ -like function” on “Solid lines are guides to the eye.”

Reviewers' comments:

Reviewer #3 (Remarks to the Author):

The reply to my question 3 sheds some light to the problem but does not make the point completely clear. Figure attached to the reply where the switching is shown in Cartesian coordinates is much clearer than that shown in the paper. The shown part of dependence is indeed well fit by $\cos(2\phi)$. I understand that beyond $+ or - 45$ deg. from the maximum the opposite domains start to grow, which can be described by negative "normalized area of relative switching" continuing the $\cos(2\phi)$ dependence. If the initial state corresponds to a demagnetized sample with $\langle M_z \rangle = 0$ and the spot size is much larger than the domain size (as it is shown in insets of Fig. 3 where the "normalized switched area" is introduced) then the "normalized area of relative switching" is proportional to the final $\langle M_z \rangle$ and probably would be perfectly described by $\cos(2\phi)$. A figure showing this quantity measured in at least 180 deg. interval (in Cartesian coordinates, of course) would be then quite clear and convincing in contrast to present Fig. 4 b. Is it indeed the case that the dependencies shown in Figs. 3, 4, and S1 are obtained for a demagnetized initial state? Would the results obtained at magnetized sample (in the sense that the domain size is larger than or compared to the laser spot size as shown in Fig 2) be identical to that? Will the "normalized switched area" be a good quantity to describe the switching efficiency in this case? I think, the manuscript should not leave these questions open and both the data presentation and the discussion should be improved in this respect.

The reply to my question 1 is also not clear. I understand that domains with the state 8 are always larger than those with the state 5 if prepared by applying external field. The question was: are there unambiguous experimental evidences that if the "1" domain with $M_z < 0$ upon the laser excitation switches into a domain with $M_z > 0$, the later is always the "8" domain but not the "5" one? Yes or No? I repeat that this question is crucial since, as far as I understand, small "5" domains are always embedded in a large "1" domains, which indicates that it is energetically favorably to transfer some part of states 1 into states 5 but not 8. Moreover, the switching diagrams in Fig. 1 show that the state 1 switches to the state 5 at considerably lower fluence than into state 8 at the measured damping parameter of 0.2. Yes, I remember the argument that this number is obtained in FMR experiments with small precession amplitude while in the switching process the amplitude is large and the damping can be larger at larger amplitudes. This however raises the question of how the model takes such amplitude-dependent damping into account since the amplitude is large in the beginning of switching and small in the end as it is evident from the trajectories in Fig.1. In this situation the conventional way could be to refer to the damping at small amplitudes introducing the necessary corrections for large amplitudes. Otherwise, the damping (which is not constant in time) is just a bad parameter to show on the axis of diagrams. If the model does not account for the amplitude-depending damping but finally the authors claim that it is dependent then it is even more severe problem: I would not suppose that some kind of "effective mean damping" can be introduced and lead to a trustable results. At least, there should be arguments for that presented. I believe that these issues must be discussed in the manuscript in more details, clear, and honestly, without hiding the thin ice under a deep snow.

I also believe that after the authors consider this discussion seriously and modify the manuscript in such a way that the above questions and issues are made clear for the reader if not solved, the publication can be granted.

REPLY TO REFEREE #3

Referee 3.1

The reply to my question 3 sheds some light to the problem but does not make the point completely clear. Figure attached to the reply where the switching is shown in Cartesian coordinates is much clearer than that shown in the paper. The shown part of dependence is indeed well fit by $\cos(2\phi)$. I understand that beyond $+ or - 45$ deg. from the maximum the opposite domains start to grow, which can be described by negative “normalized area of relative switching” continuing the $\cos(2\phi)$ dependence. If the initial state corresponds to a demagnetized sample with $\langle M_z \rangle = 0$ and the spot size is much larger than the domain size (as it is shown in insets of Fig. 3 where the “normalized switched area” is introduced) then the “normalized area of relative switching” is proportional to the final $\langle M_z \rangle$ and probably would be perfectly described by $\cos(2\phi)$. A figure showing this quantity measured in at least 180 deg. interval (in Cartesian coordinates, of course) would be then quite clear and convincing in contrast to present Fig. 4 b.

Reply: Although we disagree with the Referee, we now have changed Figure 4b (see below). As requested, it does show the very same results, but in Cartesian coordinates. For the rest, we must note that the Referee keeps coming with new requests that were not mentioned in the previous reports. In this work we only report about studies of the switching of the magnetization from state “8” to state “1”, while in previous work it has already been shown that the switching is fully reversible by the orthogonal polarization. Therefore, polarization dependences in the interval spanning more than 90-degrees do not make much sense. We added a note mentioning that we also fully verified that the switching from “8” to “1” state works as expected.

At the same time, we would like to stress that our manuscript already now represents a concise story about physics of all-optical switching in garnets which have not been even predicted before.

Is it indeed the case that the dependencies shown in Figs. 3, 4, and S1 are obtained for a demagnetized initial state? Would the results obtained at magnetized sample (in the sense that the domain size is larger than or compared to the laser spot size as shown in Fig 2) be identical to that?

Reply: Yes, they would be identical, as shown by the inset in Fig. 3. Both demagnetized state with four domains (e.g. shown in Fig. 2) and two-domain state with the same in-plane magnetization components were used to study the switching.

Studies of the switching with the spot size smaller than the size of a single stable domain, as requested by the Referee, also do not seem to be possible as the recorded domain will not be stable.

Will the “normalized switched area” be a good quantity to describe the switching efficiency in this case? I think, the manuscript should not leave these questions open and both the data presentation and the discussion should be improved in this respect.

Reply: Yes, it will be, because of the gaussian shape of the pulse - the lower the switching threshold, the larger the switched spot. In response to the criticism, we have added in the section “Methods” few more lines (lines 271-276): “This four-domain state with orthogonal in-plane magnetization components was used for demonstration of the photo-magnetic recording by scanning polarized laser beam (see Fig. 2). However for experiments of switching in domains shown in Fig. 3, Fig. 4 and Fig. S1 the demagnetization initial state with only “8” and “4” domains have been used. In these cases, the normalized switched area is a good quantity to describe the switching visualized in the magneto-optical experiment.”

Referee 3.2

The reply to my question 1 is also not clear. I understand that domains with the state 8 are always larger than those with the state 5 if prepared by applying external field.

Reply: Yes, the domain image (see the image in our reply in the previous round) shows large domains of state “8” with embedded multiple small domains of state “4”. Similarly, applying field in the opposite direction [1-10] creates large domains in state “1” with embedded small domains in state “5”.

The question was: are there unambiguous experimental evidences that if the “1” domain with $M_z < 0$ upon the laser excitation switches into a domain with $M_z > 0$, the later is always the “8” domain but not the “5” one? Yes or No? I repeat that this question is crucial since, as far as I understand, small “5” domains are always embedded in a large “1” domains, which indicates that it is energetically favorably to transfer some part of states 1 into states 5 but not 8.

Reply: Yes, the experimental identification is unambiguous, and the switching happens in the following way: under influence of a single pulse, domain “8” with embedded domains “4” switches into domain “1” with embedded domains “5”.

The procedure for identification of domains types in these samples was developed long ago and described in many publications. To avoid further misunderstandings, we attach below the list of publications devoted to studies of the domain pattern in Co-doped garnets. In all these papers as well as in our experiments the domains “1” and “8” have always been seen as large, while domains “4” and “5” have always been seen as small.

To avoid misunderstandings, we rewrite this section in “Methods”. In particular, we now write (lines 259 - 271):

“In YIG:Co thin film with small miscut angle four types of magnetic domains were observed. Any of the four domains with magnetization states 1, 4, 5 and 8 (orientations are shown in Fig.1b) could be stabilized by applying and removing an in-plane field of about $\mu_0H=2$ mT directed along one of the four easy-axes of the magnetization $\langle 110 \rangle$. The magneto-optical contrasts for “5” (or “4”) and “8” (or “1”) domains at off-normal incidence of the probe are almost the same. However, “8” and “1” - domains are always larger than “5” and “4” domains. The magnetization processes and domain structure analysis are systematically discussed in Ref. 23 and references therein. To acquire a multi-domain state as shown in Fig. 2, the sample was brought into “8” state by applying an external magnetic field of 80 mT along the [1-10] axis. After removing the field the sample turns into a state with large “8” domains and “4” small domains. Afterwards, applying 2 mT magnetic field along the [110] axis we convert a part of large “8” domains and a part of “4” domains into large “1” and small “5” domains, respectively.”

List of publications dedicated to domains in YIG:Co garnet films:

1. S.Uba, A.Maziewski, J.Simsova, Investigation of magnetic after-effect in cobalt-doped thin garnet films, *J.Phys.C: Solid State Phys.*, 16, 383 (1983).
2. A.Maziewski, M.Tekielak, P.Görnert, Magnetic after-effect in cobalt doped garnet epitaxial films, *Acta Physica Polonica A68*, 15 (1985).
3. M.Kisielewski, A.Maziewski, P.Görnert, Magnetic after-effect influence on domain-wall dynamics in Co-containing garnet films, *Appl. Phys.* 20, 222 (1987).
4. A.Maziewski, M.Kisielewski, P.Görnert, K.Brzosko, Unidirectional properties of $(YCa)_3(FeCoGe)_5O_{12}$ films, *IEEE Trans. on Magn.* MAG 23, 5, 3367 (1987).
5. A.Maziewski, B.Szymański, P.Görnert, Magnetic anisotropy in Co-doped garnet films, *Acta Physica Slovaca*, 39, 4, 232 (1989).
6. M.Kisielewski, A.Maziewski, P.Görnert, Ac susceptibility analysis of YIG+Co films, *Acta Physica Polonica A76*, 2, 283 (1989).
7. Z.Simsa, J.Simsová, P.Görnert, A.Maziewski, Absorption and Faraday rotation of YIG: Co, Ge, Ca garnet films, *Acta Physica Polonica, A76*, 2, 277 (1989).
8. A.Maziewski, V.V.Volkov, P.Görnert, Dynamics of magnetic domains in cobalt doped garnets, *Sov. Phys. Solid State*, 31, 5, 300 (1989).
9. A.Maziewski, L.Pust, P.Görnert, Magnetometrical study of cobalt doped YIG films, *Journal of Magnetism and Magnetic Materials*, 83, 87 (1990).
10. A.Maziewski, M.Kisielewski, M.Tekielak, P.Görnert, Unusual magnetic domain structure properties in YIG+Co films, *Journal of Magnetism and Magnetic Materials*, 83, 82 (1990).
11. A.Maziewski, Unexpected magnetization processes in YIG+Co films, *Journal of Magnetism and Magnetic Materials*, 88, 325 (1990).
12. A.Maziewski, E.Jackiewicz, A.Kotlicki, P.Görnert, CEMS study of the magnetic iron ions orientation in Y Ca Fe Co Ge O, *Solid State Comm. Vol.73, No.7. pp. 487-490* (1990).
13. A.Maziewski, B.Ivanov, M.Kisielewski, S.Lyakhimets, A new look on magnetic after effect, *Journal of Magnetism and Magnetic Materials*, 104-107, 361-362 (1991).
14. M.Kisielewski, O.Lichtchenko, A.Maziewski, Self-biasing effect based on mixed anisotropy idea, *Journal of Magnetism and Magnetic Materials*, 101, 213 (1991).
15. B.Ivanov, M.Kisielewski, S.Lyakhimets, A.Maziewski, Domain-wall dynamics and magnetization relaxation in magnetic materials with magnetic aftereffect, *Sov. Phys. JETP* 74 (6), 1013 (1992).
16. A.Maziewski, Domain structure stabilization in a magnet with magnetization induced anisotropy, *Ferrites Proc. ICF 6 Tokyo*, 782 (1992).
17. M.W.Werhenko, W.G.Weselago, M.Kisielewski, S.N.Lyakhimets, A.Maziewski, S.G.Rudow, U.Citko, Fotoinducirovannye, polarizacionno zavisimye izmeneniya anizotropii w ferrimagnitnyh plenkah $(YCa)(FeCoGe)O$, *Pis'ma w ZETF*, 57, 6, 352 (1993).
18. S.G.Rudov, M.V.Verchenko, V.G.Veselago, A.Maziewski, M.Tekielak, S.N.Lyakhimets, J.M.Desvignes, Photoinduced, polarization dependent changes of magnetization in Co, Ge, Ca doped YIG films, *IEEE Trans. on Magn.*, 30, 2, 791, (1994).
19. A.B.Chizhik, A.Maziewski, A.Stupakiewicz, Observation of photomagnetism in Co doped YIG films at room temperature, *Proc. ICMFS'94*, (1994).
20. A.B.Chizhik, S.N.Lyakhimets, A.Maziewski, M.Tekielak, Photoinduced magnetization switchings in cobalt doped YIG films, *JMMM*, 140 (1995).
21. A.A.Milner, N.F.Kharchenko, A.Maziewski, J.M.Desvignes, "Soft" and "rigid" photomagnetism in YIG:Co films measured by linear dichroism, *JMMM*, 140 (1995).

22. M.Kisielewski, A.Maziewski, J.M.Desvignes, Domain structure shape memory in a magnet with magnetization induced anisotropy, *J. Magn. Magn. Mat.*, 140, 1923 (1995)
23. R.Jabłoński, A.Maziewski, M.Tekielak, J.M.Desvignes, FMR study of Co-substituted yttrium iron garnet films, *J.Magn.Magn.Mat.*, 160, 367 (1996).
24. M.Tekielak, W.Andrä, A.Maziewski, J.Taubert, Magnetic transitions in cobalt-substituted yttrium iron films, *Journal de Physique IV*, 7, C1-461 (1997).
25. I.Davidenko, A.Stupakiewicz, A.Sukstanskii, A.Maziewski, Фотоиндуцированная деформация доменных границ в феррит-гранатовых пленках ЖИГ: Co, *Fizika Tverdogo Tela*, 39, 1, 1824 (1997).
26. A. B. Chizhik, I. Davidenko, A. Maziewski, A. Stupakiewicz, High temperature photomagnetism in Co doped yttrium iron garnet films, *Phys.Rev.B*, 57, 21 (1998).
27. A.Maziewski, Photomagnetic effects in cobalt doped garnet films, *Electron Technology*, 31, 1, 120 (1998).
28. I. Davidenko, A. Maziewski, A. Stupakiewicz, Linearly polarized light induced changes of domain cobalt doped YIG films, *JMMM*, 196-197, 828 (1999).
29. I. Davidenko, A. Maziewski, A. Stupakiewicz, V. Zablotskii, Description of Light Pulses Induced Changes of Magnetic Anisotropy in YIG:Co, *Materials Science Forum*, vols.373-376, 477 (2001).
30. A.Stupakiewicz, A.Maziewski, I.Davidenko, V.Zablotskii, Light-induced magnetic anisotropy in Co-doped garnet films, *Phys.Rev.B* 64, 644405 (2001).

Referee 3.3

Moreover, the switching diagrams in Fig. 1 show that the state 1 switches to the state 5 at considerably lower fluence than into state 8 at the measured damping parameter of 0.2. Yes, I remember the argument that this number is obtained in FMR experiments with small precession amplitude while in the switching process the amplitude is large and the damping can be larger at larger amplitudes. This however raises the question of how the model takes such amplitude-dependent damping into account since the amplitude is large in the beginning of switching and small in the end as it is evident from the trajectories in Fig.1. In this situation the conventional way could be to refer to the damping at small amplitudes introducing the necessary corrections for large amplitudes. Otherwise, the damping (which is not constant in time) is just a bad parameter to show on the axis of diagrams. If the model does not account for the amplitude-dependence of damping but finally the authors claim that it is dependent then it is even more severe problem: I would not suppose that some kind of “effective mean damping” can be introduced and lead to a trustable results. At least, there should be arguments for that presented. I believe that these issues must be discussed in the manuscript in more details, clear, and honestly, without hiding the thin ice under a deep snow.

Reply to Referee 3.3:

In response to the Referee criticism we would like to stress that the phase diagrams were calculated using a simple model based on the Landau-Lifshitz-Gilbert equation, where the Gilbert damping is just a constant which does not depend on the amplitude. The goal of the simulations was to show that even such a model with minimum number of parameters predicts that the symmetry of the medium does allow the switching with different polarizations. To avoid misunderstandings in the present version of the paper we emphasize that the model simulates the behavior of a single spin in the crystal lattice with the symmetry of the iron garnet, where the damping is taken as a constant.

We have added in text (lines 86-88): “This model allows to simulate the behavior of a single spin in a crystal lattice with the symmetry of the iron garnet, where the Gilbert damping is taken as a constant. The damping was treated as a fitting parameter.”